# Reshaped three-body interactions and the observation of an Efimov state in the continuum

Yaakov Yudkin [1] ✉, Roy Elbaz[1], José P. D'Incao[2,3] ✉, Paul S. Julienne [4] & Lev Khaykovich [1] ✉

Efimov trimers are exotic three-body quantum states that emerge from the different types of three-body continua in the vicinity of two-atom Feshbach resonances. In particular, as the strength of the interaction is decreased to a critical point, an Efimov state merges into the atom-dimer threshold and eventually dissociates into an unbound atom-dimer pair. Here we explore the Efimov state in the vicinity of this critical point using coherent few-body spectroscopy in $^7$Li atoms using a narrow two-body Feshbach resonance. Contrary to the expectation, we find that the $^7$Li Efimov trimer does not immediately dissociate when passing the threshold, and survives as a meta-stable state embedded in the atom-dimer continuum. We identify this behavior with a universal phenomenon related to the emergence of a repulsive interaction in the atom-dimer channel which reshapes the three-body interactions in any system characterized by a narrow Feshbach resonance. Specifically, our results shed light on the nature of $^7$Li Efimov states and provide a path to understand various puzzling phenomena associated with them.

The unique ability to fine-tune the interaction between ultracold atoms has led to the realization of a number of quantum phenomena[1], among which the Efimov effect has become a quantum workhorse that allows for the exploration of some of the deepest issues of universal few-body physics[2–5]. Near a magnetic field dependent Feshbach resonance, the strength of the interatomic interaction is characterized by the s-wave scattering length $a$, which can assume arbitrarily large values compared to the characteristic range of the interactions, i.e., the van der Walls length $r_{vdW} = (mC_6/\hbar^2)^{1/4}/2$, where $m$ is the atomic mass, and $C_6$ is the dispersion coefficient. However, not all Feshbach resonances are the same. The intricate nature of the hyperfine interactions in alkali-metal atoms allows for different couplings between the open channel and the corresponding closed channel carrying the Feshbach state. As such, a resonance is said to be broad (narrow) in case of a strong

(weak) coupling and is characterized by the dimensionless strength parameter $s_{res} \gg 1$ ($s_{res} \ll 1$)[1].

Regardless of the strength of the Feshbach resonance, the Efimov effect occurs at $|a| \to \infty$ due to the formation of an induced long-range three-body interaction of the form $-1/R^2$, where $R$ is the hyperradius[5] providing the overall size of the system. This interaction gives rise to a log-periodic series of bound Efimov states whose absolute position is determined by the short-range three-body physics (Fig. 1)[2–5]. In the case of a broad resonance, the three-body potential supporting Efimov states features a universal repulsive wall near $R \approx 2r_{vdW}$ thus preventing the atoms from probing small hyperradii. In fact, this repulsive wall is the hallmark characterizing the van der Waals (vdW) universality, according to which the ground Efimov state dissociates into the three-atom continuum at $a_-^{(0)} \approx -9.73 r_{vdW}$[6,7]. This was observed across

[1]Department of Physics, QUEST Center and Institute of Nanotechnology and Advanced Materials, Bar-Ilan University, Ramat-Gan 5290002, Israel. [2]JILA, University of Colorado and NIST, Boulder, CO 80309-0440, USA. [3]Department of Physics, University of Colorado, Boulder, CO 80309-0440, USA. [4]Joint Quantum Institute (JQI), University of Maryland and NIST, College Park, MD 20742, USA. ✉e-mail: yaakov.yudkin@gmail.com; jpdincao@jila.colorado.edu; lev.khaykovich@biu.ac.il

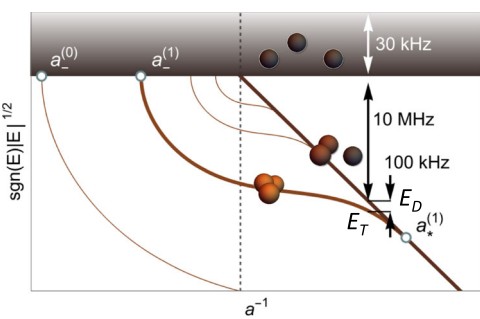

**Fig. 1 | Efimov spectrum and energy scales.** Schematic illustration of the Efimov scenario (universal theory) in the vicinity of a Feshbach resonance. The horizontal axis is the inverse scattering length $a^{-1}$, and the dashed vertical line corresponds to the position of the Feshbach resonance ($|a| = \infty$). The vertical axis indicates the wavenumber corresponding to the three-atom continuum (gray region) and the discrete spectrum of Efimov trimers (solid curved lines). The straight solid line originating at the Feshbach resonance position corresponds to the universal dimer state. The extreme points of the trimers' spectrum are labeled to indicate Efimov resonances and the first excited Efimov state is highlighted. The energy scales relevant to this work are indicated and are specific to the 894 G resonance in $^7$Li.

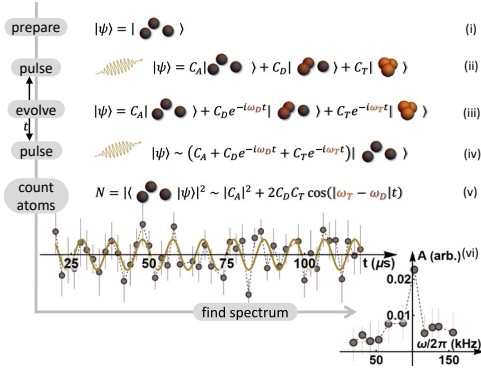

**Fig. 2 | Experimental double pulse sequence.** (i) Initial state of the three-atom continuum. (ii) A first rf pulse transfers a fraction of the initial state to dimer and trimer bound states creating a superposition state. (iii) As the wave function evolves, each constituent gains a phase proportional to its binding energy. (iv) A second rf pulse mixes the states. For simplicity, only the free-atom part is depicted. (v) Using absorption imaging, the number of free atoms is measured. (vi) An example of a measured signal as a function of free evolution time and its three-parameter fit. The signal has a large background term ($|C_A|^2$), fast oscillating terms (not shown) and a term that oscillates at $\omega = |E_T - E_D|/h$. The latter is extracted via a three-parameter fit, where $A$ indicates the amplitude.

several different Feshbach resonances in $^{133}$Cs and $^{85}$Rb[8,9]. For narrow resonances, however, this result is expected to be modified as yet another length scale emerges, namely $r_\star = 2\bar{a}/s_{\text{res}} \approx 1.912 r_{\text{vdW}}/s_{\text{res}}$. Since now $r_\star > r_{\text{vdW}}$, three-body observables are expected to depend on $r_\star$ (or equivalently, $s_{\text{res}}$) rather than $r_{\text{vdW}}$ alone[10–16]. Indeed, for intermediate resonances ($s_{\text{res}} \gtrsim 1$) deviations from the Efimov-van der Waals universality were already confirmed in recent precision measurements and calculations[17–22]. For $^7$Li atoms, although the Feshbach resonances are narrower than those above, experimental observations of $a_-$ are consistent with the vdW universality, thus challenging our understanding of universality.

Here, we show that as the resonance becomes narrower, the three-body interaction is reshaped with respect to that of a broad resonance (in any atomic species). While the universal repulsive wall near $R \approx 2r_{\text{vdW}}$ disappears, the system also develops an additional potential barrier ranging from $R \approx 4r_{\text{vdW}}$ to a distance proportional to $r_\star$, leading to a double-well structure absent for broad resonances. Specifically, we experimentally explore the energy spectrum using coherent spectroscopy in the vicinity of the atom-dimer threshold for $^7$Li atoms polarized in the $|F = 1, m_F = 0\rangle$ state, which features a Feshbach resonance at 894 G with $s_{\text{res}} \approx 0.493$, and observe an Efimov state above the atom-dimer threshold. This provides strong evidence of the reshaping for the three-body interactions for narrow resonances, and further elucidates some of the mechanisms leading to other puzzling observations with $^7$Li atoms[23–26].

## Results and discussion
### The DITRIS interferometer
In contrast to traditional cold-atom few-body experiments, which utilize inelastic losses to uncover Efimov features[27–30], we perform high-resolution coherent spectroscopy of the Efimov state on the $a > 0$ side of the Feshbach resonance. Following the proof-of-principle demonstration of ref. 31, we generate a DImer-TRImer Superposition (DITRIS) state by rf association and let it evolve in time. The accumulated relative phase between its constituents is then measured in an interferometer-like sequence. The method works best in the region around $a_\star^{(1)}$ –the value of $a$ at which the Efimov state merges with the atom-dimer threshold–where there is a clear separation of energy scales (Fig. 1). The difference between the trimer and dimer bound states must be smaller than their depth below the three-atom continuum on the one hand but larger than the temperature of the latter on the other. In this energy regime the straightforward measurement combining rf association and loss-spectroscopy fails due to rf power

broadening[30]. Our procedure thus goes beyond the existing methods. As a second condition, the rf pulse must be short enough in time such that it is Fourier broadened beyond the trimer-dimer energy difference $|E_T - E_D|$. When three atoms are subjected to such a broadband rf pulse they have the option to either form a dimer while one atom remains free or a trimer. This effectively creates a superposition of the two chemically different bound states.

The double pulse sequence is illustrated in Fig. 2. The first pulse generates DITRIS states from a fraction of a gas of free atoms. Then, following the accumulation of a relative phase according to their binding energies, the second pulse attempts to dissociate them. The dimer and trimer pathways interfere, and one observes oscillations in the number of free atoms as a function of the free evolution time. The frequency of the oscillations is proportional to $|E_T - E_D|$. The DITRIS method is thus a measurement of the trimer binding energy with respect to the atom-dimer continuum. The two requirements (separation of energies and short, Fourier-broadened pulses) set the lower and upper limits of detectable $|E_T - E_D|$. It lies between the temperature of the free-atom continuum and the pulse bandwidth, respectively (see Methods). In this regime the conversion efficiency is limited by the pulse duration, i.e. it is not saturated by the phase space density argument[32], and therefore remains low, such that $|C_A| \gg |C_D|, |C_T|$ (see Fig. 2). As a result, the oscillations appear as a small signal on top of a large background. However, making the pulses longer would decrease the upper detection limit and is therefore not favorable. To faithfully extract the main frequency contribution, we use a Fourier transform-inspired three-parameter fit (for details, see Supplementary Note 2 and 3 as well as ref. 31). As is typical for frequency measurements, the accuracy increases for longer measurements. The free evolution time is thus varied over a wide range of values (up to ~100 μs) limited only by the coherence time of the superposition state (see Methods).

### Trimer spectroscopy
Having established a reliable tool for measuring $|E_T - E_D|$ we apply the double pulse sequence for various values of the magnetic field (scattering length) with the goal of finding the point at which $E_T \to E_D$. In Fig. 3a, measurements from the DITRIS interferometer (filled circles) are represented together with the data from the previous incoherent rf association spectroscopy (open circles)[30]. At large scattering lengths,

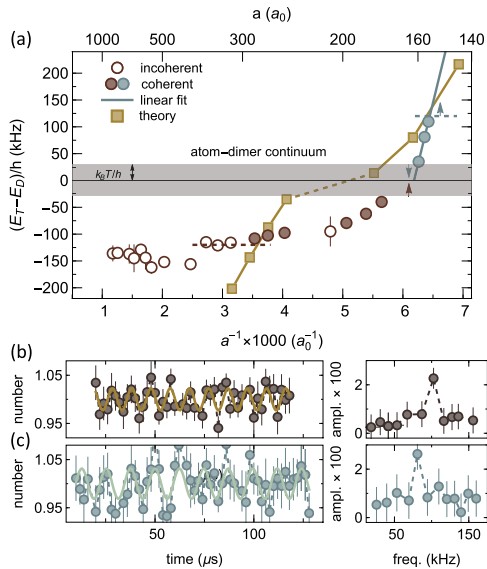

**Fig. 3 | Trimer energy from experiment and theory. a** The values of $(E_T - E_D)/h$ obtained from the double pulse sequence (filled circles) are shown together with data from rf association followed by loss (open circles)[30] as a function of inverse scattering length multiplied by 1000. For the former, the errorbars ($1\sigma$ fitting error) are smaller than the point size and so are all scattering length errorbars (see Methods). The horizontal shaded region and dashed lines show the respective lower and upper detection limits of the DITRIS interferometer. The numerical results (filled squares) for $(E_T - E_D)/h < 0$ were obtained from the methodology used in refs. 39,40 while the results for $(E_T - E_D)/h > 0$ are those extracted from Fig. 5. Being treated differently, the two regions are connected by a dashed line. **b, c** On the left panel—examples of the experimental signal (number of atoms as a function of the time between pulses). Each point is the average of 10–20 measurements and the errorbars show the standard deviation. On the right panel—results of the three-parameter fit applied to the corresponding signals which clearly indicate the presence of dominant frequencies. The horizontal (smaller than point size) and vertical errorbars are $1\sigma$ fitting errors. (For further details see Supplementary Note 2 and ref. 31). **b** For data below the threshold, at $a = 265a_0$, and **c** for data above the threshold, at $a = 156a_0$. The experimental signal and three-parameter fit for the remaining points are shown in Supplementary Fig. 3. Note that the experimental data predict the crossing of the threshold somewhere between $177\,a_0$ and $160\,a_0$.

the Efimov state is relatively deeply bound, $(E_T - E_D)/h \lesssim -100$ kHz, and our new measurements agree with those obtained from incoherent spectroscopy[30]. However, as the scattering length decreases and the Efimov state becomes more weakly bound, instead of the expected gradual approach towards the atom-dimer continuum[2–5], a sharp turn in the energy is observed. Subsequently, the experimental signal disappears for energies below the lower detection limit [see arrows in the shaded region in Fig. 3a). The latter is set by the temperature via $|E_T - E_D|/h \lesssim k_B T/h \approx 30$ kHz, due to the loss of coherence amplitude[33] (see also Supplementary Note 2). Most surprisingly, however, meaningful frequencies reemerge when the scattering length is further decreased (gray circles in Fig. 3a). The Efimov state binding energy quickly changes away from the threshold again and becomes undetectable above the higher frequency detection limit set by the pulse bandwidth [see the upper gray dashed line in Fig. 3a) leading to measurements with no dominant frequency contribution (gray arrow) (see Supplementary Fig. 3). Figure 3a also shows our theoretical results for the energies of the $^7$Li Efimov state (squares). These results, along with the physical interpretation of the phenomena controlling the observations, are discussed later in the text.

Experimentally we are only sensitive to the absolute value of the energy difference, which leads to two equally plausible scenarios: the

trimer either crosses into or bounces off the atom-dimer continuum. Although the latter scenario has been indicated in the literature to occur for broad resonances[34,35], our numerical simulations for $^7$Li instead show that the Efimov trimer crosses the atom-dimer continuum threshold due to a reshape of the three-body interaction potential associated with the narrow character of its Feshbach resonance (see discussion below).

We emphasize that the trimer remains in a metastable state well inside the atom-dimer continuum. This is demonstrated in Fig. 3b, c, where two-time sequences of the DITRIS interferometer are compared. In Fig. 3b, we show a signal obtained below the threshold and in Fig. 3c one from above it. Both signals are similar, and no observable decay is detected within the first 100 µs, covering up to 10 full oscillations. Although a thorough investigation of the trimer lifetime with the DITRIS interferometer is beyond the scope of this work, it is clear from these signals that the coherence time exceeds the expected lifetime of the Efimov trimer. (Our numerical simulations estimate the lifetime of the trimer state within the experimental range to be around 10–20 µs.) Interestingly, a recent theoretical study (performed for broad resonances) has provided a possible interpretation of such unusually large coherence times[33], with coherence still being observed for times as long as twice the lifetime of the Efimov state. Although this result does not fully explain the experimentally observed coherence times, our analysis below demonstrates fundamental differences between the three-body physics for broad and narrow resonances as $^7$Li, which can potentially lead to substantial modifications of the coherence times.

Finally, we argue the implausibility of attributing the nonzero signal from the DITRIS interferometer above the threshold to any molecular state other than the Efimov state. Although one cannot completely rule out that a non-universal (non-Efimovian) trimer state exists by accident in the same energy region in which our observations are performed, this coincidence is very unlikely. In particular, the region in phase space (above the threshold) that we explore experimentally is extremely narrow, covering only a few $a_0$ in scattering length and only a few tens of kHz in energy. Moreover, for DITRIS interferometry to provide a detectable signal, it is necessary for all states involved in the problem to be extraordinarily large. Near the region where we observe the crossing, the dimer state itself should be ~160$a_0$, and the trimer should be comparable to or even greater than that. On the other hand, a non-Efimovian accidental state could only originate from short-range physics and would be $\lesssim r_{vdW} = 32a_0$ for $^7$Li. For such small states, the coupling between them and the initial atomic state (with a size comparable to the average interatomic distance, i.e., ~$10^4 a_0$ for our case) would be extraordinarily small due to the poor Frank-Condon factor, and the DITRIS interferometer would be inefficient. As a result, since we know that the only weakly bound dimer state is the Feshbach dimer, it is reasonable to accept that the only trimer that can overlap well with both the dimer and the initial atomic state is an Efimov state.

## Theory and numerical simulations

In order to better understand the nature of the $^7$Li Efimov trimer near the atom-dimer threshold, we have performed numerical calculations using the adiabatic hyperspherical representation (see Methods). In the following, we first present a two-channel interaction model with variable $s_{res}$. This model gives insight into the crucial difference between broad and narrow resonances in the context of three-body Efimov interactions. Building upon the physical picture that emerges from the two-channel model, we then develop a multichannel theory using realistic $^7$Li two-body potentials. This latter model qualitatively reproduces the trimer's crossing of the atom-dimer threshold, thus verifying the experimental observations. We note that while necessary approximations in our theoretical model hinder quantitative agreement with the experiment, our findings clearly identify the physical mechanism controlling the experimental observations.

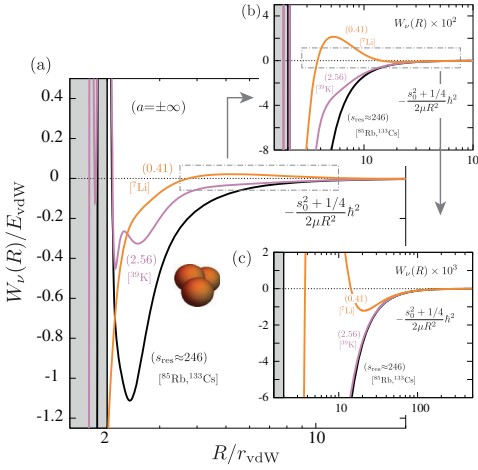

**Fig. 4 | Reshape of three-body interactions for narrow resonances. a** Effective potentials, $W_\nu(R)$, for the relevant channel supporting an infinity of Efimov states for different values of $s_{res}$ in units of $E_{vdW} = \hbar^2/mr_{vdW}^2$. As $s_{res}$ evolves from the regime of broad ($s_{res} \gg 1$) to narrow ($s_{res} \ll 1$) resonances a repulsive interaction emerges for $R \gtrsim 4r_{vdW}$ and extends up to $R \approx 3r_\star$, where $r_\star \approx 1.912 r_{vdW}/s_{res}$. The double-well structure of the three-body interaction for narrow resonances allows for trimer states to exist above the atom-dimer continuum for finite values of $a > 0$ as shape resonances. **b** 100-fold zoom of the dot-dashed box in **a**. Likewise, **c** shows a 10-fold zoom of the dot-dashed box in **b**. For more values of $s_{res}$ see Supplementary Fig. 5.

## Three-body interactions near narrow resonances

The two-channel model we use for the interatomic interaction contains the proper van der Waals physics and a set of parameters chosen to produce a Feshbach resonance with the $^7$Li background scattering length, $a_{bg} \approx -25a_0$[36], but variable values for $s_{res}$ (see Supplementary Note 4). In the adiabatic hyperspherical representation, a great deal of physical insight can be obtained from the hyperspherical effective potentials $W_\nu(R)$, which are solutions of the adiabatic Hamiltonian at fixed values of the hyperradius $R$. In Fig. 4 (see also Supplementary Fig. 5), we show the effective potentials relevant for Efimov physics at $a = \pm\infty$ and various values of $s_{res}$ between 0.41 and 246, thus covering the broad, intermediate and narrow resonance regimes. Asymptotically, all potentials approach the universal form $-(s_0^2 + 1/4)\hbar^2/2\mu R^2$ with $s_0 \approx 1.00624$, which supports infinitely many Efimov states. However, at shorter distances the potentials are drastically reshaped as the resonance strength enters the narrow resonance regime. (We note that similar results have been found in a recent publication[16]).

For our broadest resonance ($s_{res} \approx 246$), representing atomic species like $^{85}$Rb and $^{133}$Cs, the effective potential displays the expected universal repulsive wall near $R \approx 2r_{vdW}$ (thick black curve in Fig. 4a), which prevents atoms from probing the small $R$ region representing the hallmark of vdW universality[6,7]. As $s_{res}$ is tuned towards the intermediate ($s_{res} \approx 2.56$, similar to $^{39}$K) and narrow ($s_{res} \approx 0.41$, similar to $^7$Li) resonance regime, the effective potentials are reshaped, and the universal repulsive wall eventually disappears. Remarkably, the three-body potentials develop a repulsive barrier for $R \gtrsim 4r_{vdW}$ (Fig. 4b), which extends up to $R \approx 3r_\star$ (Fig. 4c), as a result of the strong mixing between the open and closed hyperspherical channels (see Supplementary Note 4). Therefore, in the $s_{res} < 1$ regime, the effective potentials display a double-well structure, where interactions within the inner well ($R \lesssim 4r_{vdW}$) are dominated by vdW interactions while interactions in the outer well ($R \gtrsim 3r_\star$) are dominated by Efimov physics. For the $s_{res} \approx 0.41$ case, the closest to $^7$Li ($s_{res} \approx 0.493$), the potential barrier height is found to be ~10 MHz ($0.02E_{vdW}$) at $a = \pm\infty$, i.e. much larger than the range of binding energies found experimentally. Importantly,

and relevant to our present experiment, this barrier also persists for finite values of $a > 0$ (see Supplementary Note 4).

## Multichannel calculations for $^7$Li

To provide a more quantitative analysis of the effect of the repulsive barrier in the parameter regime of the experiment, we perform additional numerical calculations that characterize the energy of the $^7$Li Efimov state using a more realistic interaction model based on the methodology developed in refs. 18,19. We note that modifications to this model were made (see Supplementary Note 4) in order to compensate for strong (short-ranged) electronic exchange interactions[37,38]. Yet, our model displays the correct physics at distances $R \gtrsim r_{vdW}$, thus preserving the major features relevant to the central physical question we explore here, i.e., whether the repulsive barrier allows for Efimov states to exist above the atom-dimer threshold.

While our results for $(E_T - E_D)/h < 0$ in Fig. 3a were obtained using a methodology that provides a direct characterization of the energy of the Efimov state[39,40], for $(E_T - E_D)/h > 0$, the analysis of the near-resonant energy regime in the atom-dimer continuum is much more subtle. However, a convenient way to characterize the existence of the $^7$Li Efimov state above the atom-dimer threshold is to compare the energy dependence of the $^7$Li atom-dimer elastic cross-section, $\sigma_{AD}$, with that of a system without the barrier, i.e., a system controlled by a broad ($s_{res} = \infty$) Feshbach resonance, $\sigma_{AD}^\infty$[35]. It is crucial, however, that this comparison is performed when the two physical systems have the same value for the atom-dimer scattering lengths, $a_{AD} = a_{AD}^\infty$, such that both cross-sections converge to the same value, $4\pi|a_{AD}|^2$, as the collision energy vanishes. In this case, if $a_{AD} < 0$ and an Efimov state exists above the threshold, the cross-section difference should display the enhancement whenever the collision energy coincides with that of the $^7$Li Efimov state. In practice, the above procedure is most meaningful in the case of weak inelastic transitions, which has led us to suppress the short-range decay mechanisms in our model. Also, since, in our case, the values of $a_{AD}$ we obtained are only approximately the same (differing by no more than 2%), we define the weighted cross-section difference as

$$\Delta\sigma_{AD}^w(E) = \sigma_{AD}(E) - \frac{|a_{AD}|^2}{|a_{AD}^\infty|^2}\sigma_{AD}^\infty(E), \qquad (1)$$

which ensures that the cross-section difference vanishes as the collision energy $E \to 0$. In the above expression, $\sigma_{AD}(E) = \pi|1 - S_{AD}(E)|^2/k_{AD}^2$ where $S_{AD}$ is the diagonal $S$-matrix element associated with the atom-dimer channel and $k_{AD}^2 = 2m/3E/\hbar^2$, with $m$ being the atomic mass. The results for $\Delta\sigma_{AD}^w$ for various $a_{AD}$ in Fig. 5 clearly show the expected enhanced scattering of $^7$Li with respect to the broad resonance case thus demonstrating the existence of a $^7$Li Efimov state above the threshold as a direct consequence of the existence of the repulsive barrier. The energy of the Efimov state is associated with the maximum value of $\Delta\sigma_{AD}^w$ occurring at smaller values of $E$ as $|a_{AD}|$ increases, i.e., when the state approaches the atom-dimer threshold from above, and is displayed in Fig. 3a. We note that simplified, asymptotic models have failed to explain our experimental observations, indicating the importance of van der Waals interactions in order to properly describe the reshape of the three-body interactions[41,42]. Note also that, as shown in Fig. 3a, when $(E_T - E_D)/h < 0$, the theory results predict deeper energies close to the Feshbach resonance, which dive faster towards the atom-dimer continuum. We attribute such discrepancies to the simplifications adopted for otherwise nearly intractable, truly multi-channel interactions in lithium. Even the most advanced attempt to model these interactions[38] have not reached fully converged results, leading to a significant discrepancy between theory and experiment[24,25] for the spin state considered here. Most importantly, however, is that both our experimental data and theoretical simulations refute the

conventional expectation that an Efimov state simply merges with the atom-dimer continuum for the case of narrow resonances.

In summary, our experimental and theoretical observations of the existence of an Efimov state above the atom-dimer continuum provides strong evidence of a fundamental reshaping of the three-body interactions for narrow resonances. Although our theoretical analysis allows us to point out that this phenomenon is universally valid for narrow Feshbach resonances, much is still needed to fully characterize the $^7$Li Efimov states, in particular, with respect to their lifetime. The coherence times observed with the DITRIS interferometer are clearly much longer than the estimations of the trimer lifetime obtained from our numerical simulations without coherence. Since we show here that for $^7$Li atom-dimer collisions are enhanced, this raises the intriguing question on whether the character of the DITRIS superposition state, along with the form of the three-body interaction, can conspire to form more long-lived superposition states in a way that coherence can still be observed at long times[33]. Although some open questions still remain, our current observations provide evidence that Efimov physics at a narrow Feshbach resonance deviates from the expectations from vdW universality, where the Efimov state simply disappears at the atom-dimer threshold.

Successful application of the DITRIS interferomter to coherent spectroscopy of the Efimov energy level, together with a notable demonstration of coherent manipulation of $^4$He halo dimers by ultra-short laser pulses[43], reveals the great potential of the coherent approach to few-body physics phenomena. Future investigation into the superposition state lifetime, and the extension of the unique capabilities of the DITRIS interferometer to other atomic species and mixtures are expected to greatly advance our understanding of Efimov physics in ultracold atoms.

## Methods

### Experimental details

Standard laser cooling and evaporation techniques are used to produce a gas of $3 \times 10^4$ bosonic lithium atoms at 1.5 µK and an average density of $1.25 \times 10^{12}$ cm$^{-3}$ in a crossed optical dipole trap. The temperature corresponds to 30 kHz and is our lower detection limit. To image the atoms we use absorption imaging which is sensitive to free atoms only.

At the core of the experiment lies the 10 µs pulse, which is Fourier broadened to address both the dimer and the trimer simultaneously. The duration 10 µs refers to the full-width at half-maximum (FWHM). There is also a (measured) turn-on/turn-off time of $\tau_0 = 4$ µs, which means that the pulse is at its maximal value during $\tau_c = 6$ µs. The experimental rf pulse envelope is modeled as

$$\chi(t) = \begin{cases} \sin^2\left(\frac{\pi}{2}\frac{t + \tau_c/2 + \tau_0}{\tau_0}\right) & -\tau_0 - \frac{\tau_c}{2} < t < -\frac{\tau_c}{2} \\ 1 & -\frac{\tau_c}{2} < t < \frac{\tau_c}{2} \\ \sin^2\left(\frac{\pi}{2}\frac{t - \tau_c/2 - \tau_0}{\tau_0}\right) & \frac{\tau_c}{2} < t < \frac{\tau_c}{2} + \tau_0 \end{cases} \quad (2)$$

and plotted in Fig. 6. The Fourier transform of $\chi(t)$ is also shown in Fig. 6 in the low-frequency domain. It closely resembles a sinc, the transform of an ideal rectangular pulse, but is slightly broadened. It is FWHM is 117 kHz which is the value we use as our upper detection limit.

The method was refined with respect to the proof-of-principle in ref. 31 mainly by reducing the atom number fluctuations. This was achieved by stabilizing the magnetic field to a relative stability of $5 \times 10^{-5}$ and by improved statistics (Supplementary Note 2).

### Scattering length calibration

In Fig. 3a, the values of $(E_T - E_D)/h$ are shown as a function of inverse scattering length. In the experiment we vary the magnetic field bias to tune the scattering length. The calibration is performed via the dimer binding energy, which is frequently measured during the DITRIS interferometer data accusation. The measurement protocol can be found elsewhere[44]. Finally, the dimer binding energy is related to the scattering length via coupled channel calculations[45]. Given the high accuracy of the dimer binding energy measurement and coupled channels calculations, the scattering length uncertainty is $<a_0$ in the region explored in the experiment.

### DITRIS coherence time

The longest feasible free evolution time is given by the decoherence of the superposition state[31]. The possible relevant parameters are the elastic collision rate and the trimer's intrinsic lifetime. The low signal-to-noise ratio does not permit precise measurement of the decohrence

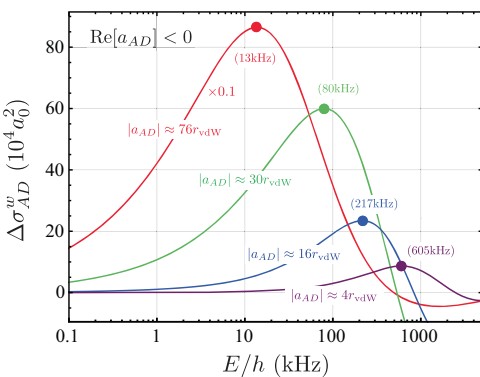

**Fig. 5 | Evidence of resonant scattering above atom-dimer threshold.** Weighted elastic cross-section difference [Eq. (1)] between $^7$Li atom-dimer collisions and that of a system controlled by a broad Feshbach resonance for approximately the same value for $|a_{AD}|$ and Re[$a_{AD}$] < 0. The existence of the repulsive barrier on the entrance atom-dimer channel for $^7$Li leads to the enhancement of elastic collisions just above the threshold as compared to that without the barrier, indicating the presence of the Efimov state above the atom-dimer threshold and with energy indicated by the closed circles. Note that at higher energies $\Delta\sigma^w_{AD} < 0$, most likely due to the fact that $|a_{AD}|$ and $|a^\infty_{AD}|$ are only approximately the same, but also due to other multichannel effects causing modifications on the scattering at such energies.

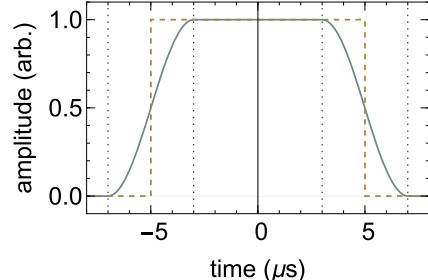
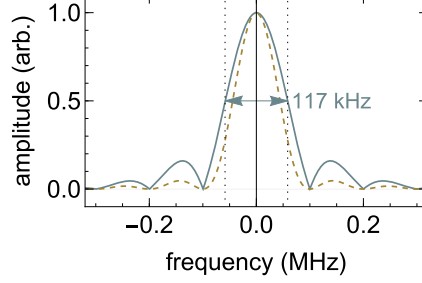

**Fig. 6 | Pulse shape and spectrum.** The pulse shape of Eq. (2) is compared to a pure square pulse with the same FWHM. The FFT of the former is slightly broader.

time but empirically we do not observe signs of decay for <150 μs. In practice, the three-parameter fit allows neglecting the decay in the data analysis by keeping the range of the evolution time <150 μs.

## Theory

The adiabatic hyperspherical representation provides a simple and conceptually clear description of the three-body system in terms of the hyperradius $R$, characterizing the overall size of the system, and the set of hyperangles, $\Omega$[46]. Bound and scattering properties[47] of the system are determined from solutions of the hyperradial Schrödinger equation:

$$\left[-\frac{\hbar^2}{2\mu}\frac{d^2}{dR^2} + W_\nu(R) - E\right]F_\nu(R) + \sum_{\nu'\neq\nu} W_{\nu\nu'}(R)F_{\nu'}(R) = 0. \tag{3}$$

where $\mu$ is the three-body reduced mass, $E$ the total energy, $W_\nu$ is the hyperspherical effective potentials governing the radial motion and $W_{\nu\nu'}$ the nonadiabatic couplings driving transitions between different channels, characterized by the collective index $\nu$. The hyperspherical effective potentials are defined as

$$W_\nu(R) = U_\nu(R) + \left\langle \Phi_\nu(R;\Omega)\left|\frac{d^2}{dR^2}\right|\Phi_\nu(R;\Omega)\right\rangle, \tag{4}$$

where $U_\nu(R)$, the hyperspherical potentials, and $\Phi_\nu(R;\Omega)$, the channel functions, are solutions of the hyperangular adiabatic equation

$$\hat{H}_{ad}\Phi_\nu(R;\Omega) = U_\nu(R)\Phi_\nu(R;\Omega), \tag{5}$$

obtained at fixed values of $R$. The adiabatic Hamiltonian contains the hyperangular kinetic energy, as well as all the atomic and interatomic interactions in the system. We explicitly define the terms of the adiabatic Hamiltonian used in our studies in Supplementary Note 4.

## Data availability

Two-dimensional raw atomic cloud pictures from all experimental runs are available upon request to Y.Y. or L.K. Source data are provided with this paper.

## Code availability

All code supporting the findings of this article and its Supplementary Information will be made available upon request to the authors.

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

## Acknowledgements

This work is supported by the US National Science Foundation (NSF, Grant no. PHY-2012125 and PHY-2308791), and the US-Israel Binational Science Foundation (BSF, Grant No. 2019795 and No. 2022740). L.K. also acknowledges the Israel Science Foundation (ISF, Grant no. 1543/20) and J.P.D. acknowledges partial support from NASA/JPL (Grant No. 1502690). This work utilized the RMACC Summit supercomputer, which is supported by the National Science Foundation (awards ACI-1532235 and ACI-1532236), the University of Colorado Boulder, and Colorado State University. The authors also acknowledge further support from the NSF grant PHY-1748958 via the Kavli Institute for Theoretical Physics.

## Author contributions

Y.Y. and R.E. performed the experiments and analyzed the data. J.P.D. and P.S.J. developed the theoretical models and performed the numerical calculations. L.K. supervised the project. All authors contributed to the discussion of the results and writing of the manuscript.

## Competing interests

The authors declare no competing interests.
