## [Peer Review File · Nature Communications]

Reshaped Three-Body Interactions and the Observation of an Efimov State in the ContinuumREVIEWER COMMENTS

Reviewer #1 (Remarks to the Author):

In this work, the authors study Efimov physics occurring in lithium-7 using a newly developed experimental method based on coherent rf drive accessing dimer and trimer states. They show evidence that the trimer state survives crossing with the atom-dimer threshold. They conjecture that this is due to the character of the underlying Feshbach resonance which has rather small pole strength s_{res} , making the physics nonuniversal. This is supported by numerical calculations performed in hyperspherical representation.

The work is quite interesting for a few-body specialist and will definitely make impact on the field of Efimov-related phenomena. It provides state-of-the-art results both on experimental and theoretical level, and the physical picture suggested here looks convincing. On the other hand, the DITRIS method has been introduced before. While I lean towards recommending the paper for publication in Nature Communications, I am not completely convinced that the method will be transferable to other systems and I am wondering about its potential to improve our understanding of few-body physics.

Other minor comments:

- 1) The Supplementary material and discussion of the fitting procedures is very helpful, as looking at the signal (e.g. in Fig. 3b) I was wondering about the quality of the fit. A reassuring sentence in the main text may be added.
- 2) At which scattering length should the trimer merge (or cross) with the atom-dimer continuum?
- 3) Can the authors provide a short summary of unusually large coherence times suggested in ref. [34]?
- 4) Fig. 4 is rather complicated, maybe a simpler one with less curves would fit better here, while this one can be moved to Supplementary Material?
- 5) On the other hand, this Figure shows the resonant case, while the most interesting feature is expected for finite values of a . How does the potential look there?
- 6) Fig. 5 and the analysis of this part is much less clear to the reader, I frankly speaking got lost in the assumptions made here.
- 7) Would decreasing the atomic temperature below 1.5 μK improve the applicability of the method? (it seems to me this sets a bound on observable $E_{\text{T-E}_D}$)

Reviewer #2 (Remarks to the Author):

The manuscript by Yaakov et al. explores the Efimov state near a narrow Feshbach resonance. In the experimental segment, the authors employ high-resolution coherent spectroscopy to discern the energy difference between the dimer and the Efimov states, with a specific focus on situations where the system is proximate to the atom-dimer threshold. Remarkably, they detect a clear oscillation signal even within the atom-dimer continuum. This constitutes novel findings that contribute valuable insights to the exploration of van der Waals universality. Given these aspects, the manuscript merits suitable publication.

Nevertheless, the authors fall short of presenting a compelling theoretical explanation. Although the two-channel calculation reveals a reshaped effective interaction potential near a narrow Feshbach resonance, giving rise to two repulsive barriers and connecting with the persistence of the Efimov state in the atom-dimer continuum, this alone cannot serve as proof or a comprehensive explanation for the experimental observations. The availability of a solution to the Schrödinger equation in the atom-dimer continuum, considering the obtained effective potential, is left unclear.

In summary, I do not recommend publication in the current form. I would reconsider this stance if the authors furnish a more convincing theoretical understanding of their experimental observations

Reviewer #3 (Remarks to the Author):

The manuscript explores the binding energy of an Efimov trimer in the vicinity of a narrow two-body Feshbach resonance. Using few-body spectroscopy of Li atoms they observe the reappearance of a bound state above the atom-dimer continuum which they claim is an Efimov trimer occupying a metastable state embedded in the atom-dimer continuum. Supporting theoretical calculations predict the reshaping of the three-body interactions for narrow resonances like the one explored in this study. They report that the repulsive barrier produced in this reshaping may support metastable Efimov states above the atom-dimer continuum.

My major concern with the manuscript is that the measurements reported do not provide strong enough evidence to support their claim that they have observed an Efimov state in the atom-dimer continuum. Below, I list some of my concerns:

1. The indirect evidence for the trimer appears to be the three datapoints measured around ~ 160 a0 in Figure 3. What evidence do they have that this measurement of the energy difference is for the energy difference between a trimer and an atom-dimer and not some other bound states?

a. From the supplementary material it appears the magnetic moment of the reappeared state is not the same as the trimer state "above the continuum"

b. Do they also observe any difference in the coherence time? Is there some characteristic of a metastable trimer that they could measure (like a shorter lifetime?)

2. The DITRIS technique appears to be a useful technique for probing these energy scales but has yet to be widely employed in other experiments. It seems there are still many open questions on the measured signals which I feel leaves the results open to some interpretation. The unexpectedly long coherence times and the role of temperature are two causes for concern over their interpretation. I explain each below:

a. Coherence times: The fact the coherence times are longer than the lifetime of the trimer casts doubt in my mind that the signals stem from trimer states. At points in the manuscript the authors state that the choice of measurement time is limited by the coherence time of the superposition but at no point do they state what this is. Did they not measure for long enough to measure this? What prevented doing so? In the supplemental material I see that at the end of section C they state that "In our experiment the sample length is ultimately limited by the decay of the signal which is $> 200\mu\text{s}$ ". Which suggests some form of measurement was performed.

b. Temperature: What is the role of temperature for these measurements? The effect of thermal broadening is mentioned with regard to the spectroscopy but temperature can play an important role in the identification of Efimov resonances. It seems from reading Ref 34 the observed signals in experiment may come only from atoms occupying the lowest motional state of the trap, though this is not commented on at all in this manuscript.

3. It is hard to pick apart how much of the theory is being modified in order to try and reproduce the experiment and if the final theoretical results are consistent with the experiment. Could the theoretical predictions be plotted along with Figure 3 to get a sense of the agreement? It seems the agreement requires a certain value of the atom-dimer scattering length to match the experiments, is this value known? Does it agree with the experimental measurements?

I also have a few further comments with the aim of improving the clarity of the manuscript, particularly for non-experts.

1. In figure 1 multiple additions to the caption (and possibly figure) could be made to improve clarity
 - a. The dashed line through the middle is not commented on but I assume this is to indicate zero? Then everything to the left is negative a and the right positive a ?
 - b. The x and y axes are not described in the caption
 - c. I think adding labels for E_t and E_d would help reader comprehension (you already have provided approximate energy scales for E_t and $(E_t - E_d)$)
2. Figure 2: I found the caption confusing to read as the label for each step (i,ii etc) was placed at the end of each sentence. It would make more sense to have the label at the start of the sentence so its easier to follow which step each sentence is describing.
3. Figure 3:
 - a. x axis of (a) is not described in the caption. To avoid confusion it may be best to explicitly say that the x labels have been multiplied by 1000 (otherwise it looks like the labels are 4000 a_0^{-1} , 5000 a_0^{-1} etc).
 - b. The apparent deviation from expected behaviour could be highlighted more clearly by showing a theory line with $E_t - E_d$ tending towards zero as described in the text (and plotted to some degree in Figure 1).
4. Pg 3 end of Para 2: Where Reference 34 is cited it would be convenient to briefly summarise the possible explanation for the coherence times. Which from my reading of the reference is from an increased Efimov lifetime and the observed signal being from excitation of pairs from the trap ground state (which I think is important to mention).
5. Figure 4: The plots either have no y label or my guess is that the y label is placed next to the label (a). In any case the y axis is not described in the caption.
 - a. It would be much clearer to label the y axis in the usual location and describe it in the caption.
 - b. I think the subplots b and c are enhanced views of the small energy regions to show the necessary effects. To make this more obvious to the reader I would put a box around the region of (a) which has been enhanced or at least mention that it is an enhanced view in the caption.

Aside from my minor comments on the presentation of the figures, the manuscript is well written. The manuscript offers some important new results in the vicinity of a narrow Feshbach resonance, exploring the behaviour of Efimov trimers near the atom-dimer threshold. The work builds upon established work exploring Efimov physics near broad Feshbach resonances and is important for probing the limits of Efimov universality. The experimental results offer the first pieces of evidence of deviation from van der Waals universality in ^7Li and the theoretical calculations offer some qualitative insight into how reshaping of the three-body interactions could accommodate Efimov trimers in the atom-dimer continuum. The significance of these results make them worthy for publication in Nature

communications. However, the experimental evidence for Efimov trimers in the continuum is indirect and this indirect evidence is not supported by quantitative agreement with the theory. Therefore, I cannot recommend publication in its current form.

I recommend that authors revise the claims of their work to better match the results or to provide more compelling evidence that their measurements are conclusively of a metastable Efimov trimer embedded in the atom-dimer continuum.

REPLY TO THE REVIEWERS COMMENTS

The reshape of three-body interactions: Observation of the survival of an Efimov state in the atom-dimer continuum

Yaakov Yudkin¹, Roy Elbaz¹, José P. D’Incao^{2,3}, Paul S. Julienne⁴, and Lev Khaykovich¹

¹*Department of Physics, QUEST Center and Institute of Nanotechnology and Advanced Materials, Bar-Ilan University, Ramat-Gan 5290002, Israel*

²*JILA, University of Colorado and NIST, Boulder, Colorado 80309-0440, USA*

³*Department of Physics, University of Colorado, Boulder, Colorado 80309-0440, USA and*

⁴*Joint Quantum Institute (JQI), University of Maryland and NIST, College Park, Maryland 20742, USA*

(Dated: December 29, 2023)

We thank all three referees for their thorough reading of our manuscript and corresponding comments and suggestions. Our general impression is that their assessment is quite positive. However, the referees also point to important aspects that cause them concerns. We believe our revised manuscript addresses all such concerns and hope the referees will agree that this version of our manuscript is suitable for publication in Nature Communications. Below, we address the referees comments in detail and indicate the corresponding changes in the text. For clarity, we copy the referees’ report and added our reply highlighted in red.

I. REVIEWER #1 (REMARKS TO THE AUTHOR):

In this work, the authors study Efimov physics occurring in lithium-7 using a newly developed experimental method based on coherent rf drive accessing dimer and trimer states. They show evidence that the trimer state survives crossing with the atom-dimer threshold. They conjecture that this is due to the character of the underlying Feshbach resonance which has rather small pole strength s_{res} , making the physics nonuniversal. This is supported by numerical calculations performed in hyperspherical representation.

The work is quite interesting for a few-body specialist and will definitely make impact on the field of Efimov-related phenomena. It provides state-of-the-art results both on experimental and theoretical level, and the physical picture suggested here looks convincing. On the other hand, the DITRIS method has been introduced before. While I lean towards recommending the paper for publication in Nature Communications, I am not completely convinced that the method will be transferable to other systems and I am wondering about its potential to improve our understanding of few-body physics.

We thank the referee for the positive assessment of our work. We emphasize that it is not our intent to introduce the DITRIS method here, but rather show its crucial role in uncovering a strikingly new and unexpected few-body physics phenomenon, namely, observation of the survival of the Efimov state in the atom-dimer continuum. This experimental discovery has triggered an in-depth theoretical investigation which further revealed fundamental modifications of the three-body interactions near narrow Feshbach resonance which are responsible for the observations. Therefore, we believe the present manuscript should be viewed as a first demonstration of the full strength of DITRIS interferometer method in revealing new few-body physics phenomena.

Regarding the referee’s comment on the applicability of DITRIS interferometry, we emphasize that there is no fundamental limitation to creating a DITRIS with atomic species other than lithium. For example, Ref. [35] of the current version considers the DITRIS scenario for ⁸⁵Rb atoms and shows that oscillations can be observed under reasonable experimental conditions. Extension to other species is expected to be rather straightforward. In the revised manuscript we have modified the paragraph starting with “Having established a reliable tool for measuring...” to better emphasize the limits of DITRIS regarding the temperature and pulse length, thus allowing the reader to evaluate the relevant experimental parameters for other atomic species.

Besides its broad appeal, we also believe that the DITRIS interferometer has a great potential to enrich future research on few-body interactions. First, it allows for detailed study of the lifetime of the DITRIS state (provided the SNR is improved) which is currently one of the most intriguing open questions in the field. Also, coherent measurements with the DITRIS interferometer will allow for the investigation of *elastic* collisions between dimers and atoms by studying either collisional narrowing or broadening effects as a function of atom density. So far, no other method has been able to characterize *elastic* atom-dimer collisions, an important aspect to understand strongly interacting atom-dimer quantum mixtures and Efimov physics. Second, the DITRIS method allows for an improved (more precise) measurement of the atom-dimer Efimov resonance position using a different experimental approach as compared to the standard measurement of inelastic losses in the atom-dimer mixture. Comparison between the results of these two different approaches can provide better understanding of such few-body phenomena. It would be

especially interesting to realise the DITRIS method in the vicinity of other narrow Feshbach resonances to improve our knowledge of the three-body interactions in this regime.

There are simply too many prospective studies with the DITRIS interferometer to describe them all in the manuscript. However, motivated by the referee's comments we revised our manuscript to better reflect our view on the DITRIS method and the relevance of our findings. First we clarify significance of current studies as following:

- In the abstract we slightly changed the two last sentences to emphasize universality of our findings, i.e. the metastable trimer state should appear in *any* system governed by a narrow Feshbach resonance: “We identify this behavior with a novel universal phenomena related to the emergence of a repulsive interaction in the atom-dimer channel which reshapes the three-body interactions in any system characterized by a narrow Feshbach resonance. Specifically, our results shed new light on the nature of ^7Li Efimov states and provide a new path to understand various puzzling phenomena observed here, as well as in other previous experimental studies”.
- In the second paragraph of the manuscript we explicitly emphasize that the reshape of the three-body interaction is applied universally to any atomic species and depends exclusively on the strength of the resonance pole: “Here, we show that as the resonance becomes narrower (in any atomic species), the three-body interaction is reshaped with respect to that of a broad resonance.”
- In the conclusion paragraph we rephrase the last sentence to emphasize again a general significance of our discovery: “Although some open questions still remain, our current observations provide the first evidence that Efimov physics at a narrow Feshbach resonance deviates from the expectations from vdW universality where the Efimov state simply disappear at the atom-dimer threshold.”

Second, we added the last paragraph to the manuscript to emphasize promising research directions where the DITRIS interferometer impact is expected to be crucial:

- “Successful application of the DITRIS interferomter to coherent spectroscopy of the Efimov energy level together with a notable demonstration of coherent manipulation of ^4He halo dimers by ultrashort laser pulses [39] reveals a great potential of the coherent approach to few-body physics phenomena. Future investigation into the superposition state lifetime, and the extension of the unique capabilities of the DITRIS interferometer to other atomic species and mixtures are expected to greatly advance our understanding of Efimov physics in ultracold atoms.”

Other minor comments:

1) The Supplementary material and discussion of the fitting procedures is very helpful, as looking at the signal (e.g. in Fig. 3b) I was wondering about the quality of the fit. A reassuring sentence in the main text may be added.

When such a signal appears for the first time on Fig. 2 we indeed comment on data analysis and refer the reader to the Supplemental Material. To attract the reader's attention to this point even more we enhanced the caption of Fig. 3 when referring to subplots (b) and (c): “(b),(c) On the left panel - examples of the experimental signal (number of atoms as a function of time between pulses), and on the right panel - results of the three-parameter fit applied to the corresponding signals which clearly indicate the presence of preferred frequencies (for details see Supplemental Material and Ref. [6])” In addition, we refer the reader to the Supplemental Figure with all the experimental data.

2) At which scattering length should the trimer merge (or cross) with the atom-dimer continuum?

To indicate the crossing scattering length we added the following sentence at the end of Fig. 3 caption: “Note that the experimental data predicts the crossing of the threshold somewhere between $177 a_0$ and $160 a_0$ ”.

3) Can the authors provide a short summary of unusually large coherence times suggested in ref. [34]?

We thank the referee for this suggestion and we have addressed it accordingly in our revised manuscript. The study in Ref. [34] ([35] of our revised manuscript) has shown that the interferometric scheme like ours can produce coherence times as long as twice the lifetime of the Efimov states. This occurs simply because the main decay mechanism relevant for coherence originates from the cross terms between the three-body states forming the DITRIS state. See the discussion regarding Eq. (7) of Ref. [34]. Although this longer coherence time does not provide a quantitative explanation of our observed coherent times, it is also important to emphasize that such theoretical studies were performed for a system possessing a broad Feshbach resonance, not a narrow one as in our case. As our present study shows, the three-body physics for narrow resonances is quantitatively different, and in general will strongly affect atom-dimer interactions in ways not captured in Ref. [34] and can potentially enhance the coherence lifetimes. We have added some of this discussion in our revised manuscript. See the last sentence of the paragraph starting with “Experimentally we are only...”

4) Fig. 4 is rather complicated, maybe a simpler one with less curves would fit better here, while this one can be moved to Supplementary Material?

We have followed the referee's suggestion and replace the original Fig. 4 with a simpler version and moved the original to the supplementary material. The simpler version of Fig. 4 now only includes cases relevant for atomic species explored experimentally.

5) On the other hand, this Figure shows the resonant case, while the most interesting feature is expected for finite values of a . How does the potential look there?

We thank the referee for this comment. We have added in the supplementary material a figure that shows the three-body potentials for a finite value of $a > 0$. This new figure illustrates how the repulsive barrier evolves from the resonant ($a = \pm\infty$) case, as suggested by the referee.

6) Fig. 5 and the analysis of this part is much less clear to the reader, I frankly speaking got lost in the assumptions made here.

We thank the referee for bringing out attention to this point. We have rewritten the corresponding paragraph discussing Fig. 5. We hope the referee will find it more clear. We also have updated Fig. 5 with some additional calculations that better emphasizes the enhancement of collisions as the Efimov state is closer to the atom-dimer threshold.

7) Would decreasing the atomic temperature below 1.5 μ K improve the applicability of the method? (it seems to me this sets a bound on observable $E_T - E_D$)

Yes, indeed, we expect that lower temperature can potentially improve the resolution limit of DITRIS interferometer. Finite temperature effects are suppressed in the regime where the binding energy is larger than the temperature. [See discussions in Ref. [34] (now Ref. [35]).] Nevertheless, a more fundamental limit will be set by the finite lifetime of the superposition state. According to this consideration (see Ref. [6]), the smallest energy we could measure would be most probably set at about 10 kHz (a factor of 3 lower than reported in this work) which would require lower temperatures than we currently have. However, another limitation has to be taken into account when lowering the temperature which is related to the BEC threshold. At lower temperatures (a bit less than 1 μ K) a BEC will be formed with an unavoidable increase in density. This can potentially decrease the lifetimes and, ultimately, degrade the resolution limit. For this reason, we have avoided working in such a regime. We hope, however, the changes we have made in the revised manuscript will provide to the reader a better view on the role of the temperature, as well as other factors, to our measurements.

II. REVIEWER #2 (REMARKS TO THE AUTHOR):

The manuscript by Yaakov et al. explores the Efimov state near a narrow Feshbach resonance. In the experimental segment, the authors employ high-resolution coherent spectroscopy to discern the energy difference between the dimer and the Efimov states, with a specific focus on situations where the system is proximate to the atom-dimer threshold. Remarkably, they detect a clear oscillation signal even within the atom-dimer continuum. This constitutes novel findings that contribute valuable insights to the exploration of van der Waals universality. Given these aspects, the manuscript merits suitable publication.

Nevertheless, the authors fall short of presenting a compelling theoretical explanation. Although the two-channel calculation reveals a reshaped effective interaction potential near a narrow Feshbach resonance, giving rise to two repulsive barriers and connecting with the persistence of the Efimov state in the atom-dimer continuum, this alone cannot serve as proof or a comprehensive explanation for the experimental observations. The availability of a solution to the Schrodinger equation in the atom-dimer continuum, considering the obtained effective potential, is left unclear. In summary, I do not recommend publication in the current form. I would reconsider this stance if the authors furnish a more convincing theoretical understanding of their experimental observations.

We thank the referee for the positive assessment of our experimental findings and their corresponding relevance to the understanding of van der Waals universality. We also appreciate the referee's more critical view on how our theoretical studies were presented. We have made substantial changes to the theory part and have established a much closer connection to the experimental results. We hope the referee will find that this clarifies their important points that were left unclear in our original manuscript.

To specifically address the referee's comments, we want first to emphasize that we use the two-channel model simply as a tool to illustrate the reshape of the three-body interactions as we move from the broad to the narrow resonance regime. This is the most conceptually clear approach to understand the physical differences between both such regimes. Even though our findings using the two-channel model are themselves important, we evidently agree with the referee that they alone do not provide a convincing explanation to the experimental data. For this reason, we have also performed numerical calculations using a more realistic model for ^7Li atoms based on the same methodology we used in Refs. [23,24], containing both the correct electronic and hyperfine structure of the system. Using this model, we explore the solutions of the Schrodinger equation below and above in the atom-dimer threshold, but, perhaps, have failed in making this point clear. Motivated by the referee's criticism we have revised the theory presentation to better

reflect the role of the theory in interpreting the experimental data. For instance, Fig. 3(a) now contains our calculated energies of the Efimov state both below and above the atom-dimer threshold. We also have better emphasized that while the procedure to calculate the energy of the Efimov state below the atom-dimer threshold is well-known and straightforward, characterizing the energies of the Efimov states above the threshold is more challenging and requires a different procedure as the one we described in our manuscript. As the referee will find in our revised manuscript, our theoretical results for the Efimov energies show the same qualitative behavior found experimentally, with Efimov states being bound both below and above the threshold. These results confirm our physical interpretation that the potential barrier we observed in the atom-dimer interaction is what allows the Efimov state to live above the threshold. We note that this barrier in the atom-dimer channel is present in both our calculations using the two-channel model and the more complete hyperfine model of Refs. [23,24] (shown now in the supplemental material).

We hope the referee will find that his major concerns about the theory presentation are now addressed and that our revised manuscript now deserves publication in Nature Communications.

III. REVIEWER #3 (REMARKS TO THE AUTHOR):

The manuscript explores the binding energy of an Efimov trimer in the vicinity of a narrow two-body Feshbach resonance. Using few-body spectroscopy of Li atoms they observe the reappearance of a bound state above the atom-dimer continuum which they claim is an Efimov trimer occupying a metastable state embedded in the atom-dimer continuum. Supporting theoretical calculations predict the reshaping of the three-body interactions for narrow resonances like the one explored in this study. They report that the repulsive barrier produced in this reshaping may support metastable Efimov states above the atom-dimer continuum.

My major concern with the manuscript is that the measurements reported do not provide strong enough evidence to support their claim that they have observed an Efimov state in the atom-dimer continuum. Below, I list some of my concerns:

1. The indirect evidence for the trimer appears to be the three datapoints measured around 160 a0 in Figure 3. What evidence do they have that this measurement of the energy difference is for the energy difference between a trimer and an atom-dimer and not some other bound states?

We thank the referee for bringing up this important point. Indeed, experimentally we measure three points above the continuum to support a presence of a bound state above the atom-dimer threshold. There are several arguments that allow us to rule out the possibility that some other bound states are the cause of our observations and support our conclusion that the state is indeed the Efimov state:

- For deep enough energies (below threshold) the extracted binding energy of the trimer state agrees very well with previously measured values based on a completely different experimental approach (incoherent spectroscopy [31]). Such studies in Ref. [31] well characterized the state as an Efimov state indicating that our interferometric observations in the energy region are also originated from the Efimov state. For small energies, the interferometric signal above the atom-dimer threshold appears in the immediate proximity to measurements below the threshold and these two groups can be naturally connected across the threshold. This continuity across the threshold serves as a reasonable account that the state above the threshold is still the same Efimov state.
- The character of the interferometric signal above the threshold is similar to that of below the threshold. This supports the assumption that the basic constituents of the signal remain the same, i.e., it is a trimer and an atom-dimer superposition state.
- Although one cannot completely rule out the possibility that a non-universal (non-Efimov) trimer state exists (by accident) in the same energy region, this coincidence can be simply assumed to be very unlikely. In particular, our observations below the threshold display a scattering length dependence which is very much characteristic of that of an Efimov state (see our previous analysis in Ref. [31]). Moreover, the region in the phase space (above the threshold) that we explore experimentally is extremely narrow. It covers only a few Bohr radius in the scattering length and only a few tens of kHz in energy. Having an ‘occasional’ three-body state in this narrow region by chance is very unlikely.
- In contrast to the appearance of an ‘occasional’ three-body state, the concept of crossing the threshold of the Efimov state into the atom-dimer continuum is supported by theory. We now show the theory prediction in Fig. 3 to provide a direct comparison between theory and experiment (see responses to Referee #1’s comments). It can be seen in the figure that theory and experiment both provide evidence of an Efimov state above and below the threshold. In the revised manuscript we emphasize this comparison.

- Another aspect to be considered is that all the states involved in the problem are of an extraordinarily large size. Near the region where we observe the crossing, the dimer state itself should be about $170a_0$, and the trimer should be comparable or even larger than the dimer state. On the other hand, a non-Efimov accidental state could only originate from short-range physics and should be smaller or comparable to the van der Waals length, $32a_0$ for ${}^7\text{Li}$. For such small states, the coupling between the initial atomic state (of size comparable to the average interatomic distance, i.e., about $10000a_0$ for our case) and such small molecular states would be extraordinarily small due to the poor Frank-Condon factor and DITRIS interferometry would be extremely inefficient, thus requiring much stronger pulses than what we use in our experiment. As a result, since we know with high certainty that the only weakly bound, large dimer state is the Feshbach dimer, it is only reasonable that the only trimer that can overlap well with both dimer and initial atomic state is a Efimov state.

Based on these arguments and corresponding modifications in our manuscript, we hope that the referee will agree that our study does provide a fully substantiated evidence of a Efimov state above the atom-dimer threshold.

a. From the supplementary material it appears the magnetic moment of the reappeared state is not the same as the trimer state "above the continuum"

The magnetic moment of the trimer changes as a function of the scattering length and, as discussed in the Supplemental Material section "Magnetic moment", it agrees well with the dimer+atom system below the threshold. Indeed, above the threshold it changes by about 10% as compared to the atom+dimer system. However, this is reasonable because above the threshold the atom+dimer system is unbound and if we would measure the same magnetic moment, the three-body system would be unbound. Therefore it is this change in the magnetic moment that supports our conclusion that the three-body state remains bound. This argument has been added to the Supplemental Material's section A (one sentence before last): "This change in the magnetic moment provides additional evidence of the emergence of the trimer as a bound state above the atom-dimer continuum. If μ_T remained unchanged, it would instead indicate the dissociation of the trimer state."

b. Do they also observe any difference in the coherence time? Is there some characteristic of a metastable trimer that they could measure (like a shorter lifetime?)

The issue of the trimer's lifetime in our experiment is still an open (and puzzling) question we are not currently able to address. As indicated by the studies presented in Ref. [34] (now [35]), the relationship between the coherence time and the trimer lifetime is not straightforward. In such studies, it was found that coherence persist for times as long as twice the trimer lifetime. The differences we find between the three-body interactions for narrow (as in our case) and broad (as in Ref. [34]) resonances clearly indicate that one should expect substantial changes on the coherence times due to the enhancement of the atom-dimer interactions in the ${}^7\text{Li}$ case. This will be a subject of an in-depth investigation in the near future. For our present study we only say that the evolution time kept within $100\ \mu\text{s}$, chosen because the upper limit of the coherence has been estimated in Ref [6] to be $> 200\ \mu\text{s}$. To clarify this point we revise the last sentence on page 2 (second column): "The free evolution time is thus varied over a wide range of values (up to $\sim 100\ \mu\text{s}$) limited only by the coherence time of the superposition state (see Methods)." We refer the reader to the Methods section where the DITRIS coherence time is discussed.

2. The DITRIS technique appears to be a useful technique for probing these energy scales but has yet to be widely employed in other experiments. It seems there are still many open questions on the measured signals which I feel leaves the results open to some interpretation. The unexpectedly long coherence times and the role of temperature are two causes for concern over their interpretation. I explain each below:

a. Coherence times: The fact the coherence times are longer than the lifetime of the trimer casts doubt in my mind that the signals stem from trimer states. At points in the manuscript the authors state that the choice of measurement time is limited by the coherence time of the superposition but at no point do they state what this is. Did they not measure for long enough to measure this? What prevented doing so? In the supplemental material I see that at the end of section C they state that "In our experiment the sample length is ultimately limited by the decay of the signal which is $< 200\ \mu\text{s}$ ". Which suggests some form of measurement was performed.

We have previously attempted to understand the relationship between the trimer's lifetime and the coherence time in Ref [6]. Unfortunately, due to very small SNR the measurement of the trimer's lifetime is not precise enough to provide any additional insight on how it affects the observed coherence times. In our present study we provide an additional discussion of coherence time in the Methods section and we now explicitly refer the reader to it when the issue is raised in the main text. A more precise (and independent) measurement of the trimer's lifetime can be done by measuring the atom-dimer loss rates near the Efimov resonance (see for instance Ref. [2] and the supplementary material of Phys. Rev. Lett. 119, 143401 (2017)). In that case, the resonance width should provide a more accurate estimation of the trimer lifetime. That, together with our planned schemes to enhance the SNR in our coherent experiment, will allow us to better understand the relationship between the trimer lifetime and the coherence times. Evidently, this will require a new experiment to create an ultracold atom-dimer gas mixture along with the development of a theory to study coherent effects incorporating the proper form of the lithium's three-body interaction found in our current study. For these reasons, in our present work we postpone the more detailed analysis of the lifetime and coherence

time and concentrate our attention on spectroscopic measurements instead and limit our measurements to times that are shorter than the estimated lifetime found in Ref. [6].

Nevertheless, we agree with the referee that the incompatibility between the coherence time and the corresponding lifetime of the Efimov trimer (as calculated by the theory) is puzzling. In our revised manuscript we have better emphasized that the relationship between these two time scales is not straightforward by referring to the study of Ref. [34] (now [35]) accordingly to the discussion we provided in response to the referee's comment 1b. We also but hope the referee will agree (based on our discussion of the referee's comment 1) that the observed signals can only be attributed to the Efimov state.

b. Temperature: What is the role of temperature for these measurements? The effect of thermal broadening is mentioned with regard to the spectroscopy but temperature can play an important role in the identification of Efimov resonances. It seems from reading Ref 34 the observed signals in experiment may come only from atoms occupying the lowest motional state of the trap, though this is not commented on at all in this manuscript.

We thank the referee for bringing this up. We do agree that temperature plays an important role in the identification of Efimov resonances. In typical collision experiments, for instance, where three-body recombination is measured, the effect of temperature is to shift the position of the Efimov resonance as well as broadening of the resonance feature. For interferometric studies like ours, the effect of the temperature is to potentially reduce the amplitude of the coherent oscillations observed in the atom population, thus making the determination of the corresponding frequency more challenging. This is shown, for instance, in the numerical simulations of Ref. [34] (now Ref. [35]). Such effects, however, should only undermine measurements whenever the temperature is comparable or larger than the energy we want to measure, i.e., the energy difference $|E_D - E_T|$. Although some of this information is already in the text, we apparently need to emphasize these points. As a result, we have revised the text (specifically the paragraph starting with "Having established a reliable tool for measuring...") to better reflect the role of the temperature in our experiments.

We now address the second point made by the referee regarding the study of Ref. [34] (now ref. [35]). The method used in ref [34] assumes three atoms under harmonic confinement with frequencies adjusted to produce a system whose average interatomic distances corresponding to that of the experiment, i.e., $\sim n^{-1/3}$. Although such a model (also previously used by some of us) is capable of producing qualitative results, the fictitious trap frequencies in this model will produce a system with a much stronger confinement compared to that used in experiments. As a result, one needs to be careful in order to interpret such studies. For instance, while in the model of ref. [34] the signal originates from the lowest "fictitious" atom-dimer trap state, the proper physical interpretation to experiments like ours is that the signal originates from the atom-dimer continuum, as we do in our manuscript.

3. It is hard to pick apart how much of the theory is being modified in order to try and reproduce the experiment and if the final theoretical results are consistent with the experiment. Could the theoretical predictions be plotted along with Figure 3 to get a sense of the agreement? It seems the agreement requires a certain value of the atom-dimer scattering length to match the experiments, is this value known? Does it agree with the experimental measurements?

We appreciate the referee's criticism and suggestions. The theoretical calculations for the Efimov state energy are now plotted along with the experimental data in Fig. 3 (a). We hope our revised manuscript can provide a better view on how our theoretical approach and subsequent adjustments and simplifications has been formulated. In short, there are two aspects our model was modified in order to provide a useful analysis of the experimental observations. The first is that we added (and kept fixed as a function of the B-field) a three-body force that adjusts the energy of the Efimov state to fall close to the experimental values. As explained in the manuscript, this compensates for the fact that ${}^7\text{Li}$ has strong spin-exchange interactions that are not fully incorporated in our model. The second modification to our theoretical approach is to suppress the decay of the Efimov state in order to perform analysis in the atom-dimer continuum. We also better explain this in the revised manuscript. None of these adjustments modify the long-range aspect of the Efimov physics in our theoretical approach, only the short-range physics.

I also have a few further comments with the aim of improving the clarity of the manuscript, particularly for non-experts.

1. In figure 1 multiple additions to the caption (and possibly figure) could be made to improve clarity a. The dashed line through the middle is not commented on but I assume this is to indicate zero? Then everything to the left is negative a and the right positive a? b. The x and y axes are not described in the caption c. I think adding labels for Et and Ed would help reader comprehension (you already have provided approximate energy scales for Et and (Et-Ed))

The caption of Figure 1 is now enhanced and reads as follows: "Schematic illustration of the Efimov scenario (universal theory) in the vicinity of a Feshbach resonance. The horizontal axis is the inverse scattering length a^{-1} and the dashed vertical line corresponds to the position of the Feshbach resonance ($|a| = \infty$). The vertical axis indicates the wavenumber corresponding to the three-atom continuum (grey region) and the discrete spectrum of Efimov trimers (solid curved lines). The straight solid line originated at the Feshbach resonance position corresponds to the universal dimer state. The extreme points of the trimers' spectrum are labelled to indicate Efimov resonances

and the first excited Efimov state is highlighted. The energy scales relevant to this work are indicated and are specific to the 894 G resonance in ^7Li .” In addition E_d and E_t labels appear now on Figure.

2. Figure 2: I found the caption confusing to read as the label for each step (i,ii etc) was placed at the end of each sentence. It would make more sense to have the label at the start of the sentence so its easier to follow which step each sentence is describing.

The figure caption is revised accordingly.

3. Figure 3: a. x axis of (a) is not described in the caption. To avoid confusion it may be best to explicitly say that the x labels have been multiplied by 1000 (otherwise it looks like the labels are 4000 a0-1, 5000 a0-1 etc).

The figure caption is revised accordingly.

b. The apparent deviation from expected behaviour could be highlighted more clearly by showing a theory line with $E_T - E_D$ tending towards zero as described in the text (and plotted to some degree in Figure 1).

The theory line is now added to the figure and explained in the caption.

4. Pg 3 end of Para 2: Where Reference 34 is cited it would be convenient to briefly summarise the possible explanation for the coherence times. Which from my reading of the reference is from an increased Efimov lifetime and the observed signal being from excitation of pairs from the trap ground state (which I think is important to mention).

We thank the referee for this suggestion and we have addressed it accordingly in our revised manuscript. The study in Ref. [34] ([35] of our revised manuscript) has shown that in the interferometric scheme like ours can produce coherence times as long as twice the lifetime of the Efimov states, even when thermal effects are taken into account. This occurs simply because the main decay mechanism relevant for coherence originates from the cross terms between the three-body states forming the DITRIS state. See the discussion regarding Eq. (7) of Ref. [34]. Although this longer coherence time does not provide a quantitative explanation of our observed coherent times, it is also important to emphasize that such theoretical studies were performed for a system possessing a broad Feshbach resonance, not a narrow one as in our case. As our present study shows, the three-body physics for narrow resonances is quantitatively different, and in general will strongly affect atom-dimer interactions in ways not captured in Ref. [34] and can potentially enhance the coherence lifetimes. We have added some of this discussion in our revised manuscript.

5. Figure 4: The plots either have no y label or my guess is that the y label is placed next to the label (a). In any case the y axis is not described in the caption. a. It would be much clearer to label the y axis in the usual location and describe it in the caption. b. I think the subplots b and c are enhanced views of the small energy regions to show the necessary effects. To make this more obvious to the reader I would put a box around the region of (a) which has been enhanced or at least mention that it is an enhanced view in the caption.

We thank the referee for the suggestions and have implemented them accordingly.

Aside from my minor comments on the presentation of the figures, the manuscript is well written. The manuscript offers some important new results in the vicinity of a narrow Feshbach resonance, exploring the behaviour of Efimov trimers near the atom-dimer threshold. The work builds upon established work exploring Efimov physics near broad Feshbach resonances and is important for probing the limits of Efimov universality. The experimental results offer the first pieces of evidence of deviation from van der Waals universality in ^7Li and the theoretical calculations offer some qualitative insight into how reshaping of the three-body interactions could accommodate Efimov trimers in the atom-dimer continuum. The significance of these results make them worthy for publication in Nature communications. However, the experimental evidence for Efimov trimers in the continuum is indirect and this indirect evidence is not supported by quantitative agreement with the theory. Therefore, I cannot recommend publication in its current form.

I recommend that authors revise the claims of their work to better match the results or to provide more compelling evidence that their measurements are conclusively of a metastable Efimov trimer embedded in the atom-dimer continuum.

We thank the referee for the positive assessment of our work. We believe that after a significant revision along the lines suggested by the referees our manuscript can be now accepted for publication in Nature Communications.

REVIEWER COMMENTS

Reviewer #1 (Remarks to the Author):

The authors improved the clarity of the manuscript. It now reads better and the results are properly highlighted. I am now convinced about the significance of the DITRIS technique which should find applications outside lithium systems. The theoretical part is now more coherent and even though quantitative agreement could not be reached, there are strong indications that the presented interpretation of the data is correct. The results are very promising for future studies of few-body physics. In my opinion the manuscript can be published in Nature Communications in this form.

Reviewer #2 (Remarks to the Author):

I appreciate the authors for their detailed response addressing my concerns. It appears that they have also provided comprehensive answers to questions raised by the other two reviewers. In comparison to the original version, the current manuscript provides a satisfactory theoretical explanation for their experimental results. Therefore, I recommend its publication.

Reviewer #3 (Remarks to the Author):

I would like to thank the authors for their modifications to the manuscript and their replies to all of my concerns. The significant changes to the manuscript have addressed my concerns and I recommend publication of this work after they address the following minor comments:

- The authors reply to my major concern regarding the measured energy difference being truly from a trimer and an atom-dimer was quite convincing. However, I note not all of their points are discussed in the text, specifically bullet points 3 and 5. It would be good to include these points in the text, probably when discussing the evidence for a metastable trimer in the atom-dimer continuum. This will provide further evidence for their claim.
- For the experimental data in Fig. 3 and Supp. Fig. 1 how is the x axis calibrated? I could not find any discussion of this in the main text or supplementary material. I think this should be included at the very least in the supplementary material.

- Related to above are there any error bars plotted on the x axis of Fig.3 and Supp. Fig. 1? If they are smaller than the points this should be commented in the caption as the authors have done for the y axis error bars in Fig. 3.

- I suspect the error bars on the magnetic moment (y-axis) measurement are smaller than the markers in Supp. Fig. 1 but there is no comment on this in the caption.

- Figure 4 is much clearer after incorporating the changes requested by myself and the other referees. However, the old figure containing the many curves has just been moved to Supp. Fig. 5 without fixing the labelling of the y axis or addressing my other comments regarding the presentation of this figure. Please can you revisit my previous comment and make the appropriate changes.

REPLY TO THE REVIEWERS COMMENTS (SECOND ROUND)
The reshape of three-body interactions: Observation of the survival of an Efimov state in the atom-dimer continuum

Yaakov Yudkin¹, Roy Elbaz¹, José P. D’Incao^{2,3}, Paul S. Julienne⁴, and Lev Khaykovich¹

¹*Department of Physics, QUEST Center and Institute of Nanotechnology and Advanced Materials, Bar-Ilan University, Ramat-Gan 5290002, Israel*

²*JILA, University of Colorado and NIST, Boulder, Colorado 80309-0440, USA*

³*Department of Physics, University of Colorado, Boulder, Colorado 80309-0440, USA and*

⁴*Joint Quantum Institute (JQI), University of Maryland and NIST, College Park, Maryland 20742, USA*

(Dated: January 23, 2024)

We thank all three referees for their thorough reading of our manuscript and corresponding comments and suggestions. In particular, we thank referees 1 and 2 for their assessment that the paper is ready for publication and referee 3 for the improvement suggestions. Below, we address the referee 3’s comments in detail and indicate the corresponding changes in the text. For clarity, we copy the referee’s report and added our reply highlighted in red.

I. REVIEWER #3 (REMARKS TO THE AUTHOR):

I would like to thank the authors for their modifications to the manuscript and their replies to all of my concerns. The significant changes to the manuscript have addressed my concerns and I recommend publication of this work after they address the following minor comments:

- The authors reply to my major concern regarding the measured energy difference being truly from a trimer and an atom-dimer was quite convincing. However, I note not all of their points are discussed in the text, specifically bullet points 3 and 5. It would be good to include these points in the text, probably when discussing the evidence for a metastable trimer in the atom-dimer continuum. This will provide further evidence for their claim.

We follow the suggestion of the referee and include an entire paragraph (the last paragraph of the "Trimer spectroscopy" subsection) which discusses this issue in detail. Specifically, we fully include bullet points 3 and 5 from our previous replay to the referee.

- For the experimental data in Fig. 3 and Supp. Fig. 1 how is the x axis calibrated? I could not find any discussion of this in the main text or supplementary material. I think this should be included at the very least in the supplementary material.

We introduced a new subsection in Methods called "Scattering length calibration" where we describe how it was done. We refer to this discussion in Fig. 3 and Supp. Fig. 1 captions.

- Related to above are there any error bars plotted on the x axis of Fig.3 and Supp. Fig. 1? If they are smaller than the points this should be commented in the caption as the authors have done for the y axis error bars in Fig. 3.

The errorbars on Fig. 3 and Supp. Fig. 1 are smaller than the size of the points in both horizontal and vertical directions. We explicitly added this information to both figure captions.

- I suspect the error bars on the magnetic moment (y-axis) measurement are smaller than the markers in Supp. Fig. 1 but there is no comment on this in the caption.

Yes, the referee is correct, the errorbars on the magnetic moment are smaller than marker size just because it is derived from the data of Fig. 3. We added a comment on it to the Figure caption.

- Figure 4 is much clearer after incorporating the changes requested by myself and the other referees. However, the old figure containing the many curves has just been moved to Supp. Fig. 5 without fixing the labelling of the y axis or addressing my other comments regarding the presentation of this figure. Please can you revisit my previous comment and make the appropriate changes.

Supp. Fig. 5 has been revised according to referee’s suggestions.

REVIEWERS' COMMENTS

Reviewer #3 (Remarks to the Author):

I thank the authors for amending the manuscript in response to my comments. I am happy to approve publication in Nature Communications in this form.